# Fetal and trophoblast PI3K p110α have distinct roles in regulating resource supply to the growing fetus in mice

Jorge López-Tello[1], Vicente Pérez-García[1,2], Jaspreet Khaira[1], Laura C Kusinski[1,3], Wendy N Cooper[3], Adam Andreani[1], Imogen Grant[1], Edurne Fernández de Liger[1], Brian YH Lam[3], Myriam Hemberger[2,4,5], Ionel Sandovici[1,3], Miguel Constancia[1,3], Amanda N Sferruzzi-Perri[1]*

[1]Centre for Trophoblast Research, Department of Physiology, Development and Neuroscience, University of Cambridge, Cambridge, United Kingdom; [2]Epigenetics Programme, The Babraham Institute, Cambridge, United Kingdom; [3]Metabolic Research Laboratories, MRC Metabolic Diseases Unit, Department of Obstetrics and Gynaecology, The Rosie Hospital, Cambridge, United Kingdom; [4]Department of Biochemistry and Molecular Biology, Cumming School of Medicine, University of Calgary, Calgary, Canada; [5]Department of Medical Genetics, Cumming School of Medicine, University of Calgary, Calgary, Canada

**Abstract** Studies suggest that placental nutrient supply adapts according to fetal demands. However, signaling events underlying placental adaptations remain unknown. Here we demonstrate that phosphoinositide 3-kinase p110α in the fetus and the trophoblast interplay to regulate placental nutrient supply and fetal growth. Complete loss of fetal p110α caused embryonic death, whilst heterozygous loss resulted in fetal growth restriction and impaired placental formation and nutrient transport. Loss of trophoblast p110α resulted in viable fetuses, abnormal placental development and a failure of the placenta to transport sufficient nutrients to match fetal demands for growth. Using RNA-seq we identified genes downstream of p110α in the trophoblast that are important in adapting placental phenotype. Using CRISPR/Cas9 we showed loss of p110α differentially affects gene expression in trophoblast and embryonic stem cells. Our findings reveal important, but distinct roles for p110α in the different compartments of the conceptus, which control fetal resource acquisition and growth.
DOI: https://doi.org/10.7554/eLife.45282.001

*For correspondence:
ans48@cam.ac.uk

Competing interests: The authors declare that no competing interests exist.

## Introduction

Intrauterine growth is dictated by the genetically-determined fetal drive for growth and the supply of nutrients and oxygen to the fetus. In turn, fetal substrate supply depends on the functional capacity of the placenta to transfer nutrients and oxygen from the mother to the fetus. Insufficient fetal substrate supply prevents the fetus from achieving its genetic growth potential and leads to intrauterine growth restriction, which affects up to 10% of the population and is associated with perinatal morbidity and mortality (*Baschat et al., 2007*; *Baschat and Hecher, 2004*). Studies have shown that the transport capacity of the placenta is diminished in human pregnancies associated with fetal growth restriction (*Glazier et al., 1997*; *Jansson and Powell, 2006*), suggesting that placental insufficiency may be the underlying cause of a fetus failing to achieve its genetic growth potential. However, there is also evidence that placental transport capacity may adapt to maintain the supply of resources appropriate for the growth potential of the fetus (*Sandovici et al., 2012*; *Sferruzzi-Perri and Camm, 2016*). For instance, the placental capacity to supply resources to the fetus is

higher in the lightest compared to heaviest placentas, supporting babies within the normal birth weight range (*Godfrey et al., 1998*). Placental supply capacity is also increased in newborns showing some protection to the growth-restricting effects of altitude-induced hypoxia (*Espinoza et al., 2001*; *Jackson et al., 1988*; *Moore et al., 2001*). Studies in mice have also shown that the placenta can adaptively up-regulate its transport function when the fetal demand for resources exceeds the supply capacity of the placenta, including following reduced expression of nutrient transporter or growth-regulatory genes, in small placentas within litters, and when the maternal provision of substrates is suboptimal (*Coan et al., 2008*; *Constância et al., 2005*; *Dilworth et al., 2010*; *Higgins et al., 2016*; *Sferruzzi-Perri et al., 2011*; *Sferruzzi-Perri et al., 2013b*; *Vaughan et al., 2013*; *Wyrwoll et al., 2009*). Taken together, these observations suggest that the placenta may adopt compensatory mechanisms to help meet the fetal genetic drive for growth, and that perhaps in severe cases of fetal growth restriction such adaptive attempts fail to occur or are insufficient. The signaling events which underlie placental adaptations, however, remain to be elucidated.

The phosphoinositide 3-kinases (PI3Ks) are a highly conserved family of enzymes, which are thought to have evolved to regulate growth in relation to nutrient supply (*Engelman et al., 2006*; *Kriplani et al., 2015*). In response to receptor activation, they generate lipid second messengers to initiate signaling cascades, which control important physiological processes such as cellular proliferation, metabolism, survival, polarity and membrane trafficking (*Vanhaesebroeck et al., 2010*). PI3Ks can be grouped into three major classes (I, II and III), on the basis of their substrate preference and sequence similarity. In mammals, Class I PI3Ks can be further subdivided according to the receptors to which they are coupled to. Class IA PI3Ks have received the most attention to date and they signal downstream of growth factor receptor tyrosine kinases (*Jean and Kiger, 2014*). In adult mammalian tissues, the ubiquitously expressed, Class IA isoform p110α plays a key role in mediating the growth and metabolic effects of insulin and insulin-like growth factors (IGFs) (*Foukas et al., 2006*; *Knight et al., 2006*; *Sopasakis et al., 2010*), which are major growth factors that also operate during feto-placental development (*Sferruzzi-Perri et al., 2017*; *Sferruzzi-Perri et al., 2013a*). Studies in mice show that changes in the PI3K pathway, particularly downstream of p110α are observed in placentas showing adaptive up-regulation of substrate supply to the fetus in pregnancies challenged with poor maternal environments and genetically-induced placental growth restriction (*Higgins et al., 2016*; *Sferruzzi-Perri et al., 2011*; *Sferruzzi-Perri et al., 2013b*). These observations indicate that p110α may be a part of a critical signaling cascade that integrates various environmental signals involved in adapting placental resource allocation to fetal growth.

Studies in mice have indicated that homozygous p110α deficiency, either by the complete loss of *Pik3ca* gene or by a mutation in *Pik3ca* which renders the p110α inactive (kinase dead mutation; p110α$^{D933A/D933A}$; α/α), results in embryonic death between gestation day 9.5 and 11.5 (*Bi et al., 1999*; *Foukas et al., 2006*; *Graupera et al., 2008*). When p110α activity is reduced by half, in mice carrying one copy of the kinase-dead mutation (p110α$^{D933A/+}$; α/+), fetuses are viable, but weigh 10–15% less than their wildtype littermates in late gestation (*Foukas et al., 2006*; *Sferruzzi-Perri et al., 2016*). Placental weight is also reduced in association with defects in the development of the labyrinthine transport region in α/+mutants (*Sferruzzi-Perri et al., 2016*). However, the capacity of the placenta to transfer maternal glucose and amino acid (via System A) to the fetus per unit surface area is increased in α/+mutants near term (*Sferruzzi-Perri et al., 2016*). This suggests adaptation of placental supply capacity with p110α deficiency. In the α/+mutant the whole conceptus is heterozygous for the p110α mutation (*Sferruzzi-Perri et al., 2016*) and the contribution of the fetal *versus* the trophoblast cell lineages in driving changes in placental development and adaptive responses cannot be discerned. This study therefore, sought to answer the following questions: 1) what effect does fetal *versus* trophoblast p110α deficiency have on placental phenotype and resource allocation to fetal growth? 2) does adaptive upregulation of placental transport depend on p110α signaling in fetal or trophoblast lineages? and 3) which genes downstream of p110α in the trophoblast compartment of the conceptus are implicated in adapting placental phenotype to support fetal growth in late mouse pregnancy? To achieve this, we selectively manipulated the expression of the p110α gene (*Pik3ca*) in the trophoblast and/or fetal cell lineages and assessed the impact on placental morphology, transport and transcriptome in relation to fetal growth.

## Results

### Fetal and trophoblast p110α interplay to regulate growth of the conceptus

We found PI3K p110α was highly expressed in the placental transport labyrinthine zone and the endocrine junctional zone of the mouse placenta (*Figure 1—figure supplement 1*). We then halved the p110α expression in either the trophoblast or the fetal (epiblast-derived) lineages of the conceptus by crossing mice in which exons 18–19 of the *Pik3ca* gene were flanked by LoxP sites (*Graupera et al., 2008*), to *Cyp19*Cre (*Wenzel and Leone, 2007*) and *Meox2*Cre (*Tallquist and Soriano, 2000*) mice, respectively, referred to as Het-P and Het-F throughout. These Cre lines are active in opposite compartments of the conceptus; for Het-P the *Cyp19*Cre targets expression in the trophoblast lineages of the placenta but not fetus, labyrinthine fetal capillaries or mesenchyme in the chorion (*Figure 1A and C*). In contrast, in Het-F the *Meox2*Cre deletes p110α expression in the fetus and placental labyrinthine fetal capillaries and chorionic mesenchyme but not trophoblast (*Figure 1B and C*) (further information regarding the genetic crosses can be found in *Figure 1—source data 1*).

We then compared the fetal and placental growth phenotype of the conditional Het-P and Het-F on day 19 of pregnancy (term ~20 days) to conceptuses with global heterozygous p110α deficiency achieved using the ubiquitous expressing *CMV*Cre line (*Schwenk et al., 1995*) (termed Het-U, *Figure 1C*). We found that compared to their wild-type (WT) littermates, there was no effect of heterozygous deficiency of p110α in the trophoblast on fetal or placental weight in Het-P mutants (*Figure 2A*). However, fetal and placental weights were 10–15% lower for Het-F and Het-U conceptuses (*Figure 2B and C*). The findings in Het-F and Het-U mutants are consistent with the proliferation defects observed in embryos with a deficiency in p110α (*Bi et al., 1999*; *Foukas et al., 2006*) and reveal for the first time that p110α in the embryo is important for determining the size of the placenta.

Although fetal and not trophoblast p110α deficiency reduced conceptus weight, both affected the structure of the placenta (*Figure 3A–C*). In the labyrinthine region of Het-P conceptuses, the volume of maternal blood spaces, fetal capillaries and surface area were reduced though trophoblast increased compared to their WT controls (*Figure 3A*). In Het-F conceptuses, the labyrinthine zone, fetal capillaries and trophoblast volume were decreased *versus* WT littermates (*Figure 3B*). In the ubiquitous p110α heterozygote (Het-U), the volume of the labyrinthine zone, fetal capillaries and trophoblast were reduced, surface area decreased and barrier to diffusion was greater, relative to WT littermates (*Figure 3C*). Reassuringly, the same defects in Het-U placental structure were previously observed for α/+mutants near term (*Sferruzzi-Perri et al., 2016*). The volume of the endocrine junctional zone in Het-P, Het-F or Het-U placentas was not significantly altered when compared to the respective WT controls (*Figure 3—figure supplement 1*). Taken together, these findings indicate that p110α in the fetal and trophoblast lineages of the conceptus interplay to regulate the development of the transport region in the placenta.

### Fetal and trophoblast p110α both regulate placental resource allocation to fetal growth

To assess whether morphological alterations in the placenta affect placental resource allocation to the fetus in Het-F, Het-U and Het-P, we measured the uni-directional maternal-fetal transfer of the non-metabolisable analogs of glucose ($^3$H-methyl-D glucose; MeG) and a neutral amino acid ($^{14}$C-methyl amino-isobutyric acid; MeAIB) on day 19 of pregnancy. We assessed fetal counts in relation to the estimated surface area for transport or to fetal weight, which respectively provided us with indices of the placental capacity for nutrient transfer and fetal growth relative to supply. We found that in compensation for the impaired labyrinthine development, Het-P and Het-U placentas transferred more MeG and MeAIB and Het-F more MeAIB per unit surface area compared to their respective WT littermates (*Figure 4A–C*). In agreement with these findings, mutant fetuses received either normal (MeG in all mutants and MeAIB in Het-P and Het-F) or increased amount of solutes (MeAIB in Het-U) for their size (*Figure 4D–F*). Moreover, the total fetal accumulation of these solutes was the same in the mutant and WT fetuses (*Figure 4—figure supplement 1*). These data suggest an adaptive response of both facilitated (MeG) and active (MeAIB) transport functions in placentas that are morphologically compromised by a lack of fetal or trophoblast p110α.

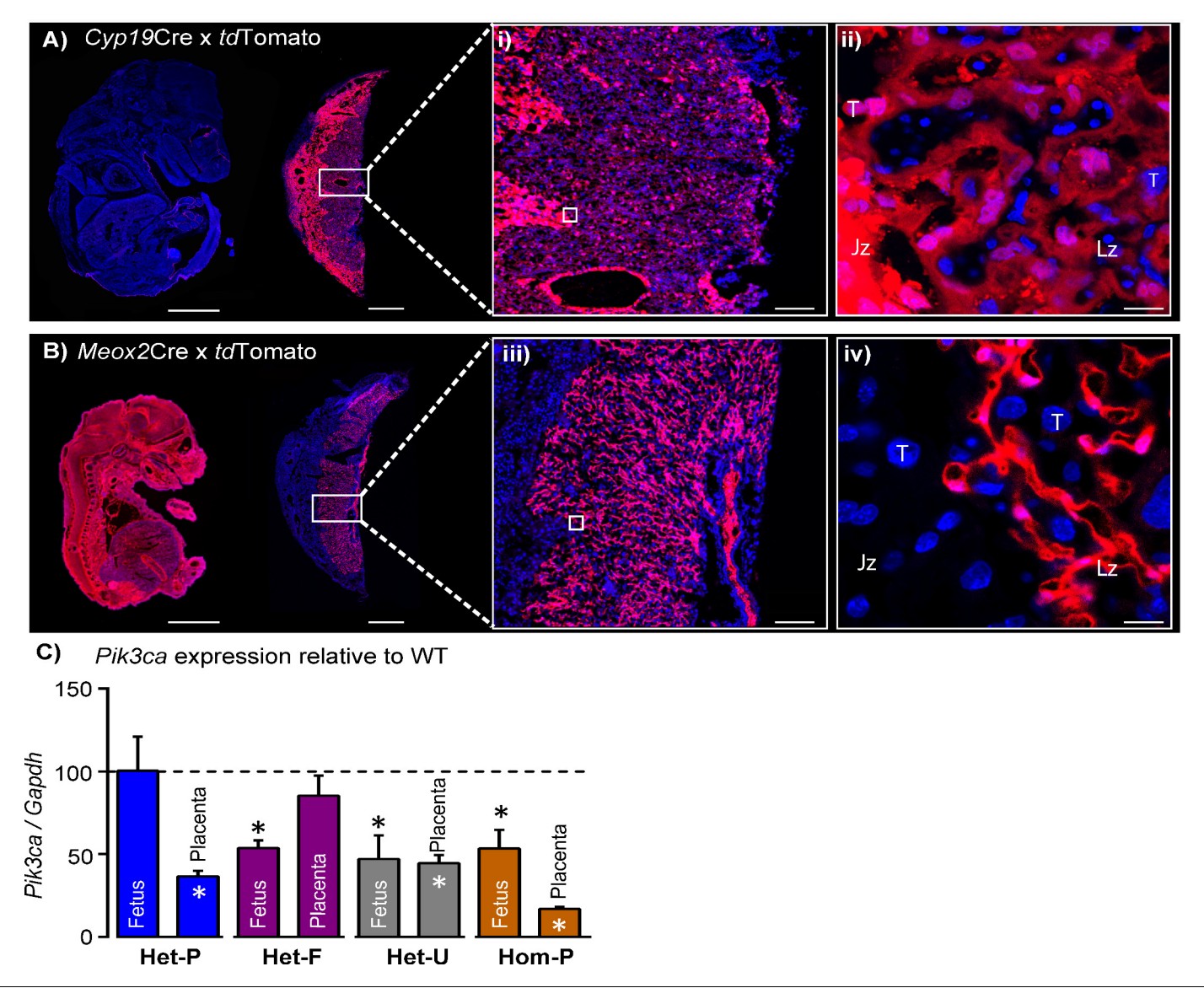

**Figure 1.** Successful modulation of p110α expression in the conceptus. (**A–B**) Validation that *Cyp19*Cre (**A**) and *Meox2*Cre (**B**) are active in opposite compartments in the conceptus by crossing lines to the *tdTomato* reporter and assessing placentas and fetuses on day 16 of pregnancy. Boxes in A and B are shown in high magnification in i and ii, respectively. Scale bar for fetuses and placentas in A and B = 2 mm and 1 mm, in i and ii = 200 μm, and in iii and iv = 20 μm, respectively. (**C**) qRT-PCR for *Pik3ca* gene normalized to *Gapdh* in whole homogenates of fetus and placenta of Het-P, Het-F, Het-U and Hom-P mutants on day 19 of pregnancy, expressed as a ratio of their respective wild-type control (WT, denoted as a dotted line). *Gapdh* expression was not affected by genotype. *p<0.05, **p<0.01, and ***p<0.001 *versus* WT, unpaired t test. n ≥ 4 per genotype. Het-F = heterozygous deficiency in the fetus, Het-P = heterozygous deficiency in the placenta, Het-U = heterozygous deficiency in the fetus and placenta, Hom-P = heterozygous deficiency in the fetus and homozygous deficiency in the placenta, Jz = Junctional zone, Lz = Labyrinth zone, T = Trophoblast.
DOI: https://doi.org/10.7554/eLife.45282.002

The following source data and figure supplement are available for figure 1:

**Source data 1.** Summary of the mouse strains and experimental crosses used in the study.
DOI: https://doi.org/10.7554/eLife.45282.004
**Figure supplement 1.** The expression of p110α protein by the mouse placenta on day 19 of pregnancy.
DOI: https://doi.org/10.7554/eLife.45282.003

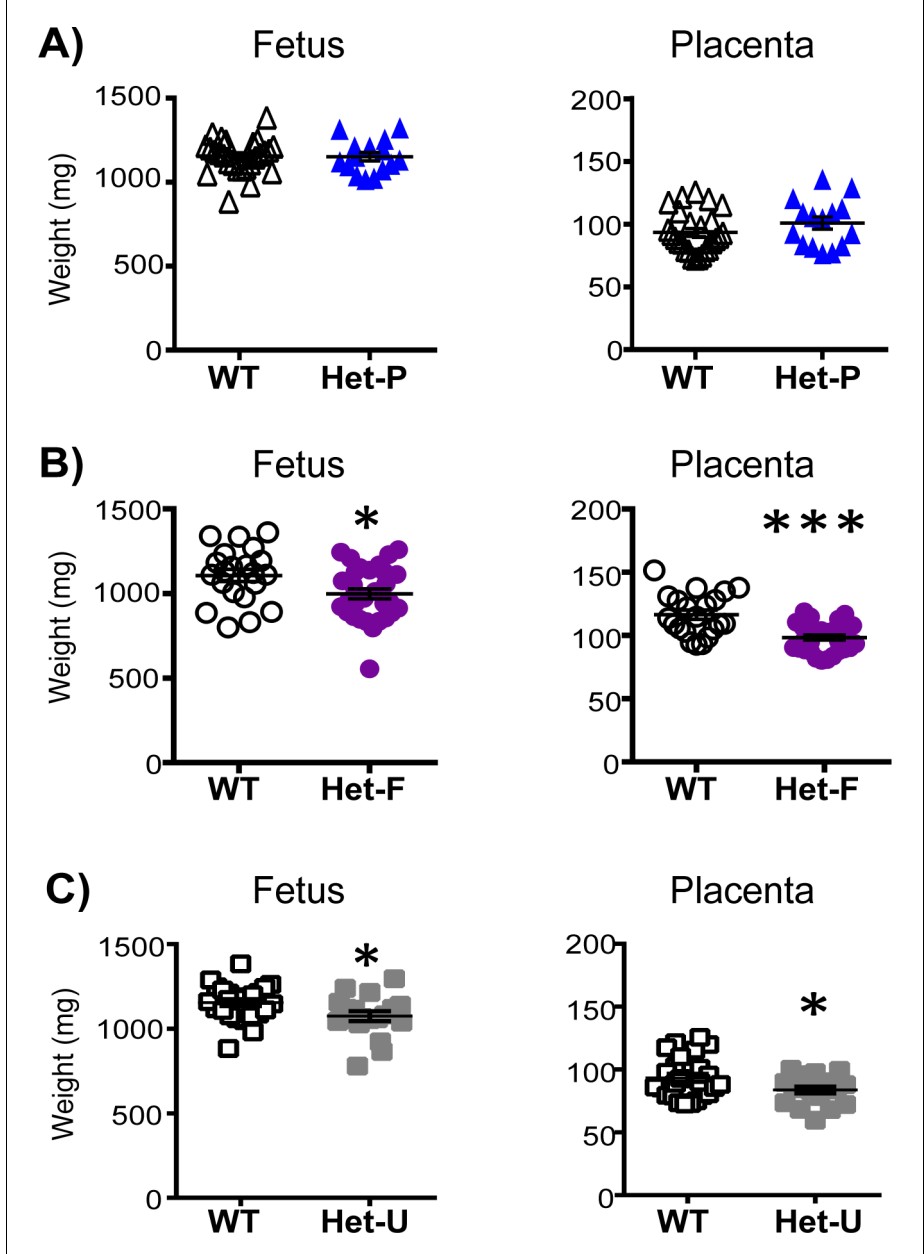

**Figure 2.** Feto-placental growth is regulated by fetal but not trophoblast p110α. (A) Het-P (n = 15) compared to WT (n = 26) (B), Het-F (n = 30) compared to WT (n = 21) and (C) Het-U (n = 18) compared to WT (n = 26) on day 19 of pregnancy. *p<0.05 and ***p<0.001 *versus* WT, unpaired t test. Note, there was no significant difference in fetal weight between Het-F and WT when data were analyzed by paired t test. Data from individual conceptuses are shown and the mean is denoted as a horizontal line with SEM.

DOI: https://doi.org/10.7554/eLife.45282.006

## Fetal p110α is essential for embryonic development and trophoblast p110α is critical for its ability to up-regulate amino acid transport to match fetal demands for growth

We wanted to know more about the regulation of placental resource allocation to the developing fetus when there is a loss of fetal and trophoblast p110α. In particular, we wondered whether adaptation of placental transport function would still occur in heterozygous mutants (Het-U) if the trophoblast or fetal lineages were completely deficient in p110α. To do this, we selectively deleted the

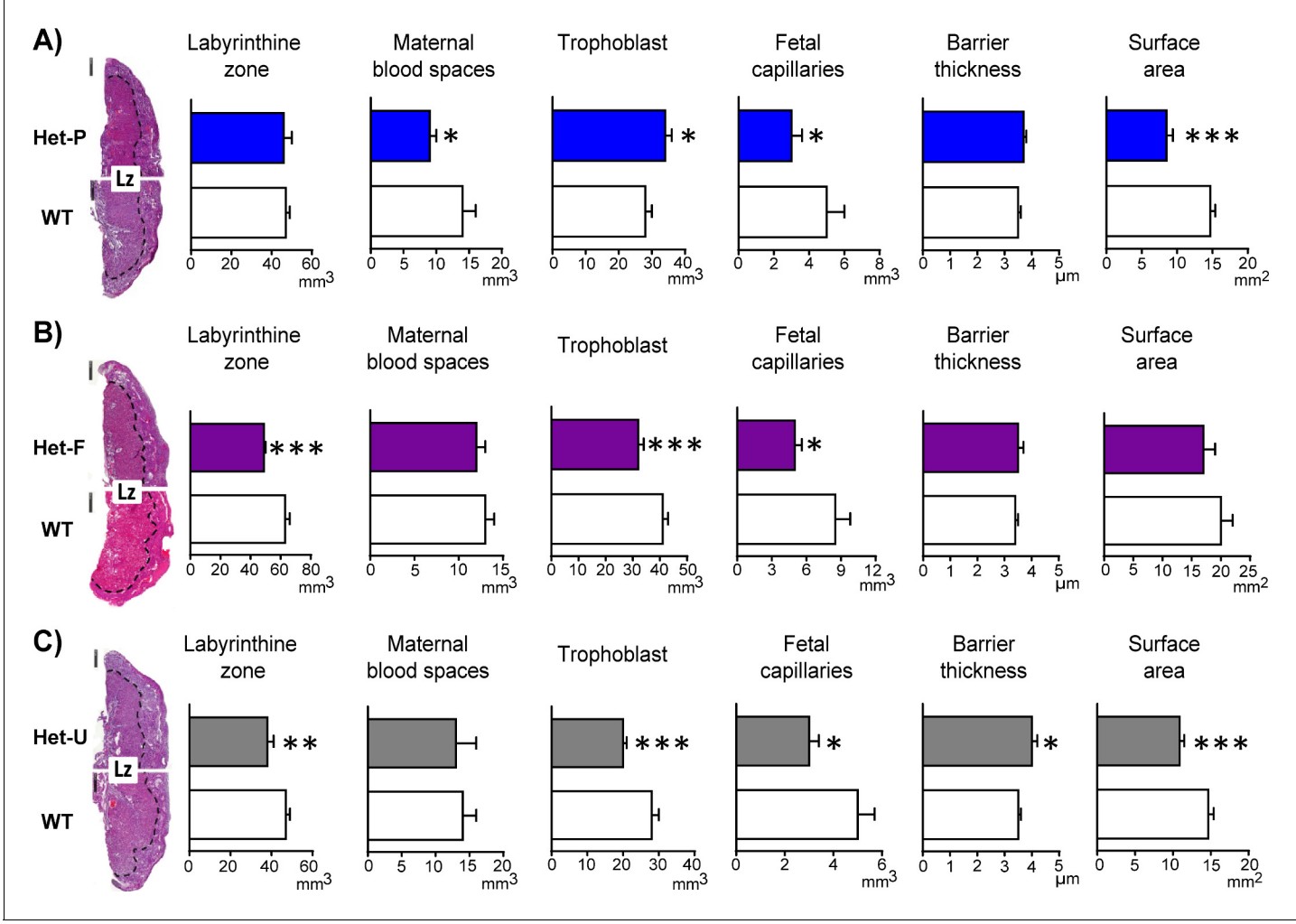

**Figure 3.** Placental morphology is regulated by both fetal and trophoblast p110α. (A) Het-P, (B) Het-F and (C) Het-U placental morphology on day 19 of pregnancy as determined by stereology. *p<0.05, **p<0.01, and ***p<0.001 *versus* WT, unpaired t test. Data presented as means ± SEM. Scale bar for hematoxylin and eosin-stained placental cross-sections is 500 μm. 6–10 placentas were assessed for each genotype for labyrinthine zone (Lz) volume and 4–5 analysed for Lz morphology.

DOI: https://doi.org/10.7554/eLife.45282.008

The following figure supplement is available for figure 3:

**Figure supplement 1.** Junctional zone volume is not modified by fetal and/or trophoblast loss of p110α.

DOI: https://doi.org/10.7554/eLife.45282.009

remaining p110α from the trophoblast or the fetal compartment of Het-U mice, using *Cyp19*Cre and *Meox2*Cre, respectively (termed Hom-P and Hom-F, respectively). We found that deleting the remaining p110α from the fetal lineages was lethal between days 11 and 12 of pregnancy (*Table 1*). In contrast to the lethality of Hom-F embryos, we found viable Hom-P fetuses in late gestation (*Figure 1C* and *Table 1—source data 1*). The timing of Hom-F lethality was identical to mutants with constitutive homozygous deficiency of p110α (α/α; *Foukas et al., 2006*) and is consistent with the role of p110α in early murine embryonic development (*Xu et al., 2009*). Taken together, these findings highlight that p110α in the fetal, but not the trophoblast compartment of the conceptus, is obligatory for prenatal development.

When comparing the growth of the Hom-P conceptuses to the control Het-U littermates, we found that fetal growth was restricted by a further 8% on day 19 of pregnancy (*Table 2*). However, despite the reduction in fetal growth, there was no difference in placental weight and labyrinthine morphology in Hom-P *versus* Het-U (*Table 2*). These observations suggest that the more severe

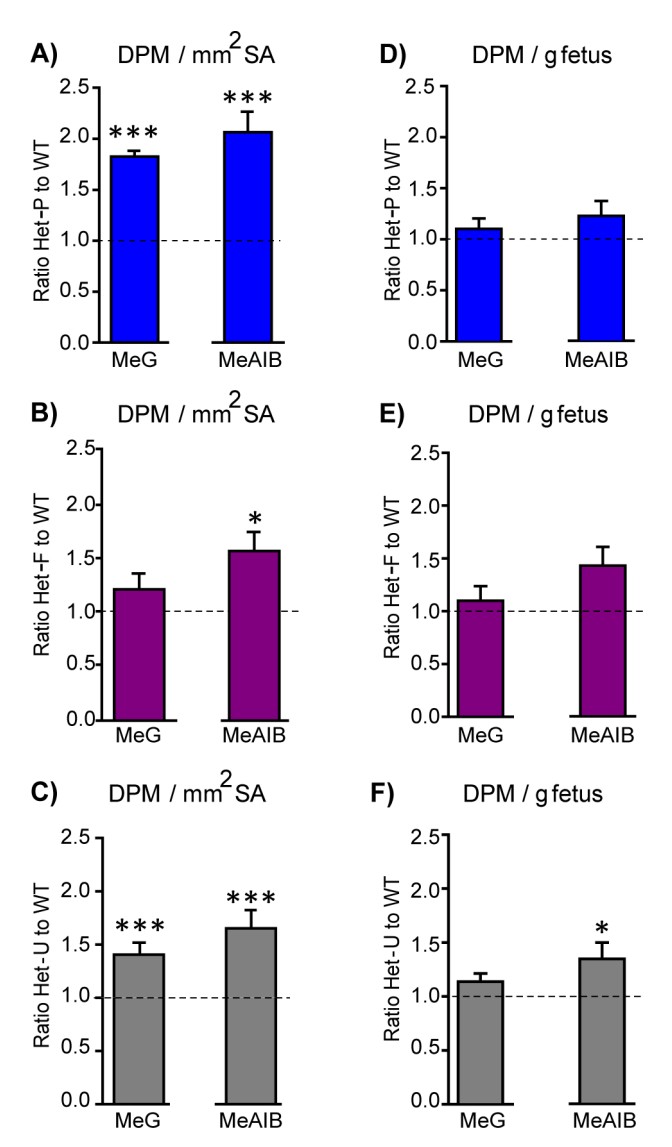

**Figure 4.** Placental nutrient transfer is adaptively increased in response to fetal and/or trophoblast loss of p110α. (A–F) The capacity of the placenta to transport $^3$H-methyl-D glucose (MeG) and $^{14}$C-amino isobutyric acid (MeAIB) relative to surface area available or to fetal weight (D–F) on day 19 of pregnancy for Het-P (n = 15) (**A, D**), Het-F (n = 24) (**B, E**) and Het-U (n = 16) (**C, F**), expressed as a ratio of their respective WT control values (denoted as a dotted line, n = 21, 18 and 21, respectively). DPM = disintegrations per minute. *p<0.05 and ***p<0.001 *versus* WT, unpaired t test. Data presented as means ± SEM.
DOI: https://doi.org/10.7554/eLife.45282.010

The following figure supplement is available for figure 4:

**Figure supplement 1.** Total solute accumulation in response to fetal and/or trophoblast loss of p110α.
DOI: https://doi.org/10.7554/eLife.45282.011

reduction in fetal growth in Hom-P (relative to Het-U), was not caused by additional defects in the formation of the placental exchange region due to a complete loss of p110α in the trophoblast.

We wondered whether the greater reduction in fetal growth may have been caused by a defect in placental transport function (a failure of the placenta to adapt its transfer capacity) in the Hom-P *versus* the Het-U mutants. We found that Hom-P placentas transferred 30% less amino acid (MeAIB) for the surface area available than Het-U littermates (*Figure 5A*). Furthermore, Hom-P fetuses received less MeAIB solute for their size, as well as overall (*Figure 5—figure supplement 1*). As the

**Table 1.** Deleting the remaining p110α from fetal compartment of the conceptus results in fetal lethality in Hom-F mutants between days 11 and 12 of pregnancy.

Frequency of viable fetuses in a litter per gestational age are displayed, with the number of litters in parentheses.

|          | WT  | Het-F | Het-U | Hom-F |
| -------- | --- | ----- | ----- | ----- |
| D10 (5)  | 23% | 26%   | 34%   | 17%   |
| D11 (6)  | 35% | 23%   | 29%   | 13%   |
| D12 (2)  | 50% | 25%   | 25%   | 0%    |
| D13 (3)  | 39% | 33%   | 28%   | 0%    |

DOI: https://doi.org/10.7554/eLife.45282.012

The following source data is available for Table 1:

**Source data 1.** Deleting the remaining p110α from the trophoblast in Hom-P does not affect fetal viability at day 19 of pregnancy.

Frequency of viable fetuses in a litter are displayed, with data from n = 15 litters. Offspring genotypes were determined by conventional PCR, and in the case of *Cyp19*Cre mutants, additionally by qRT-PCR to identify those with a sufficient level of *Pik3ca* deletion (frequency is in parentheses). When the cut off for *Pik3ca* deletion in the placenta using qRT-PCR was applied ($<65\%$ for Het-P and $<30\%$ for Hom-P), the frequency of *Cyp19*Cre mutants was ~50% less.

DOI: https://doi.org/10.7554/eLife.45282.005

Het-U placenta up-regulated its transport capacity (*Figure 4C*), this suggests that the greater reduction in fetal growth was due to an inability of the Hom-P placenta to adaptively increase amino acid transfer to the fetus. Placental glucose transfer capacity however, was not affected by a lack of placental p110α; transfer of MeG by the Hom-P placenta was equal to Het-U (*Figure 5—figure supplement 1*). Data on feto-placental growth and placental transfer in Hom-P compared to wild-type and Het-P, which in contrast, retain p110α in fetus, are shown in *Table 2—source data 1* and *Figure 5—source data 1*. Collectively, our data suggest that the demand signals of the compromised feto-placental unit for more amino acids operate via p110α in the trophoblast.

**Table 2.** Deleting the remaining p110α from the trophoblast reduces fetal weight but does not alter placental growth on day 19 of pregnancy, in Hom-P relative to Het-U controls.

Conceptus weights are from $n \geq 18$, Lz and Jz volume from $n \geq 6$ and Lz morphology from $n \geq 4$ per genotype. Data are presented as means ± SEM. * *versus* Het-U, $p<0.05$, unpaired t test.

|                                   | Het-U       | Hom-P        |
| --------------------------------- | ----------- | ------------ |
| Fetus (mg)                        | 1075 ± 31   | 994 ± 22*    |
| Placenta (mg)                     | 84 ± 3      | 89 ± 3       |
| Labyrinthine zone (mm$^3$)        | 38 ± 3      | 36 ± 2       |
| Junctional zone (mm$^3$)          | 31 ± 3      | 30 ± 2       |
| Maternal blood spaces (mm$^3$)    | 13 ± 3      | 11 ± 1       |
| Trophoblast (mm$^3$)              | 20 ± 1      | 22 ± 2       |
| Fetal capillaries (mm$^3$)        | 3 ± 1       | 4 ± 1        |
| Barrier thickness (μm)            | 4.0 ± 0.2   | 3.9 ± 0.1    |
| Surface area (mm$^2$)             | 11 ± 1      | 10 ± 1       |

DOI: https://doi.org/10.7554/eLife.45282.013

The following source data is available for Table 2:

**Source data 1.** The effect of deleting the remaining p110α from the trophoblast in Hom-P on feto-placental growth relative to WT and Het-P.

Hom-P * *versus* WT or † *versus* Het-P. *$p<0.05$ and ***$p<0.001$, †$p<0.05$ and †††$p<0.001$, unpaired t test. Conceptus weights are from $n \geq 15$, Lz and Jz volume from $n \geq 6$ and Lz morphology from $n \geq 4$ per genotype on day 19 of pregnancy. Data are presented as means ± SEM.

DOI: https://doi.org/10.7554/eLife.45282.007

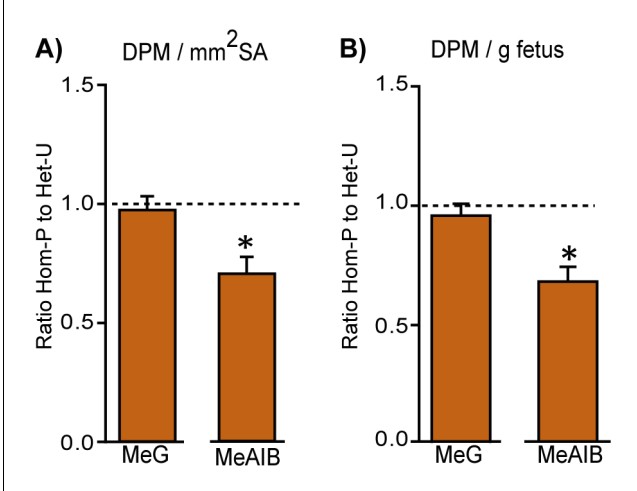

**Figure 5.** Retaining trophoblast p110α is critical for the ability of the placenta to increase amino acid transport. MeG and MeAIB transport relative to surface area available (**A**) or to fetal weight (**B**) on day 19 of pregnancy in Hom-P (n = 16), expressed as a ratio of Het-U control values (denoted as a dotted line, n = 16).
DPM = disintegrations per minute. *p<0.05 *versus* Het-U, unpaired t test.
DOI: https://doi.org/10.7554/eLife.45282.014
The following source data and figure supplement are available for figure 5:

**Source data 1.** The effect of deleting the remaining p110α from trophoblast in Hom-P on placental transport capacity relative to WT and Het-P.
DOI: https://doi.org/10.7554/eLife.45282.016
**Figure supplement 1.** Total solute accumulation in Het-U versus Hom-P.
DOI: https://doi.org/10.7554/eLife.45282.015

## Several genes downstream of p110α in the trophoblast are implicated in changes in placental phenotype to support the growing fetus

To identify genes responsible for the phenotypic differences observed between Het-U and Hom-P placentas at day 19 of pregnancy we compared their transcriptome using RNA-seq. We identified 97 differentially expressed genes, with 61 up- and 36 down-regulated in Hom-P *versus* Het-U placenta (*Figure 6—source data 1*). As expected, the FPKM (fragments per kilobase of transcript per million mapped reads) for the floxed exons, 18–19 of *Pik3ca* (*Graupera et al., 2008*) was diminished for Hom-P *versus* Het-U (*Figure 6—figure supplement 1*). Using DAVID functional annotation followed by REViGO filtering we found that genes that were down-regulated in Hom-P compared with Het-U placentas have been implicated in cytolysis and cell death, proteolysis, regulation of immune effector processes, whilst those that were up-regulated have proposed roles in cellular hormone metabolic processes (*Figure 6—figure supplement 2*). We confirmed using qRT-PCR the differential expression of several granzyme encoding genes, which are implicated in apoptosis (*Gzmc*, *Gzmf*, *Gzme*, *Gzmd*, *Gzmb* and *Gzmg*) (*Figure 6—source data 1*). Granzymes, as well as perforin-1 (which was also differentially expressed in the Hom-P *versus* Het-U placenta; *Figure 6—source data 1*) are enriched in uterine natural killer cells. However, we found no significant difference in uterine natural killer cell abundance between Hom-P and Het-U placentas (*Figure 6—figure supplement 3*). Moreover, paradoxically, we found increased levels of apoptosis in the junctional zone of the Hom-P relative to the Het-U placenta (*Figure 6B and C*). Inspecting the list of genes from the RNA-seq dataset revealed that expression of the main glucose and System A amino acid transporters (*Slc2a*, *Slc38a*) and their proposed regulators (*Mtor*, *Tbc1d4*) (*Mîinea et al., 2005*; *Rosario et al., 2013*) was not different between Hom-P and Het-U placentas. However, genes involved in water and oxygen transport (eg. *Aqp1*, *Aqp5*, *Hba-a2*) were altered (*Figure 6—source data 1*), suggesting that other transport capabilities of the placenta may be altered and contribute to the greater fetal growth restriction observed in the Hom-P *versus* Het-U. Other differentially expressed genes that were not featured in the pathway analysis have been implicated in regulating placental physiology (eg, *Cdx2*,

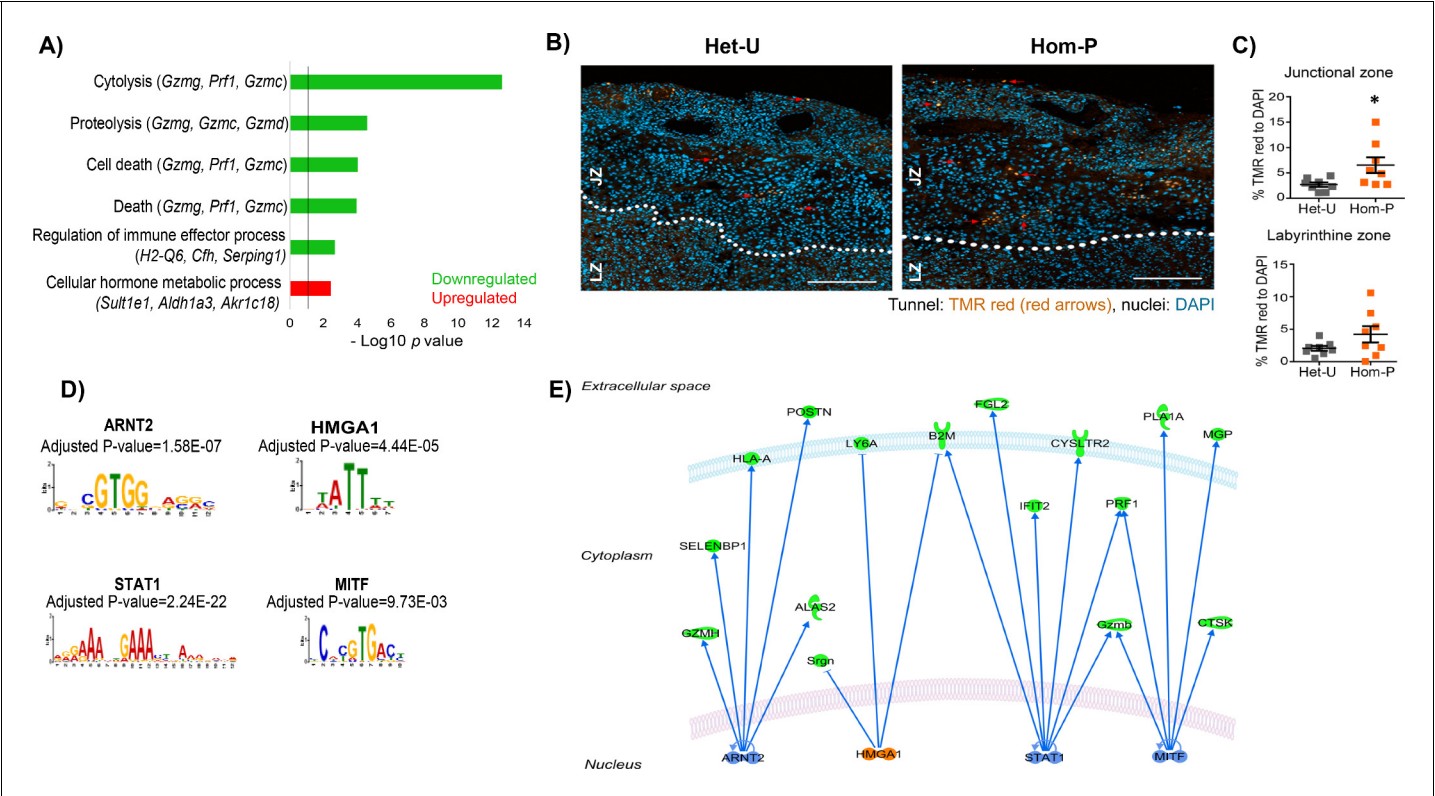

**Figure 6.** Genes downstream of p110α in the trophoblast implicated in changes in placental phenotype on day 19 of pregnancy. (A) Top-scoring biological processes enriched in Hom-P (n = 4) *versus* Het-U (n = 5) placentas on day 19 determined by RNA-seq (see also *Figure 6—source data 1*). GO terms enriched in up-regulated genes shown in red and those down-regulated in green. Three genes with the highest fold expression changes are indicated in parentheses. The line corresponds to p=0.05. (B) Representative photomicrographs of increasing magnification of cells in the placenta undergoing apoptosis in situ in Het-U and Hom-P mutants on day 19 of pregnancy. Arrows indicate cells undergoing cell death (Tunnel: TMR red, nuclei: DAPI). (C) Quantification of cells in the placenta undergoing apoptosis in situ for Het-U (n = 8) *versus* Hom-P (n = 8) on day 19 of pregnancy. Scale bar = 200 μm, Tunnel: TMR red, nuclei: DAPI, Jz = Junctional zone, Lz = labyrinth zone. *p<0.05, unpaired t test. Data presented as individual values with means ± SEM shown. (D) Transcription factors with binding sites enriched at the promoters of differentially expressed genes, as identified by Analysis of Motif Enrichment (AME). (E) Regulatory network built with the four TFs identified by AME analysis using ingenuity pathway analysis. In green are shown proteins that are down-regulated at mRNA level in Hom-P versus Het-P placentas, in blue are depicted TFs predicted as repressors and in orange TFs predicted as activators.

DOI: https://doi.org/10.7554/eLife.45282.017

The following source data and figure supplements are available for figure 6:

**Source data 1.** List of differentially expressed genes obtained by RNA-seq comparing 5 Het-U and 4 Hom-P placentas on day 19 of pregnancy.
DOI: https://doi.org/10.7554/eLife.45282.021
**Figure supplement 1.** Sashimi plot showing RNA-seq reads for exons 18-20 of *Pik3ca* for the 4 Hom-P and 5 Het-U placentas analysed on day 19 of pregnancy.
DOI: https://doi.org/10.7554/eLife.45282.018
**Figure supplement 2.** Validation of RNA-seq data.
DOI: https://doi.org/10.7554/eLife.45282.019
**Figure supplement 3.** Abundance of uterine natural killer cells in Hom-P *versus* Het-U placentas.
DOI: https://doi.org/10.7554/eLife.45282.020

*Cited2*, *Prl5a1*, *Prl7c1*, *Psg19* and *Psg22*) and pregnancy outcome (eg, *Fgl2*, *Acta2*, *Ngf* and *Nov/ Ccn3*) (*Figure 6—source data 1*).

We next identified significant enrichment of binding sites for four transcription factors (STAT1, ARNT2, HMGA1 and MITF) at the promoters of genes differentially expressed between Hom-P and Het-U placenta (*Figure 6D and E*). The mRNA expression of the transcription factors were not altered in the Hom-P versus Het-U placenta (*Figure 6—source data 1*), although previous work has shown that their activity is modified by PI3K signaling (*Mounayar et al., 2015*; *Sheu et al., 2017*;

*Terragni et al., 2011*; *Zhang et al., 2018*) and/or implicated in the development of pregnancy complications related to placental dysfunction (*Than et al., 2018*). Therefore, p110α operates via several genes in the trophoblast to alter placental phenotype to support fetal growth.

## PI3K p110α loss differentially affects the expression of genes in the trophoblast and fetal cell lineages

We wanted to know whether the genes differentially expressed (DEGs) between Hom-P and Het-U placentas, were expressed in the trophoblast or embryonic lineage of the conceptus. Using histology, we assessed the expression of a few differentially expressed genes/proteins in the Hom-P and Het-U placenta (*Acta2*/ACTA2, *Cited2*/CITED2, *Lum*/LUM). We found that these were localized to both trophoblast (in junctional and labyrinthine zones) and epiblast-derived compartments (including fetal vessels in the labyrinthine zone) in the placenta (*Figure 7—figure supplement 1*). However, only the abundance of CITED2 protein followed a similar direction to the gene, in Hom-P compared with Het-U placentas.

We also compared our RNA-seq dataset with existing trophoblast and embryonic stem cell (TS and ES cell) transcriptomes using SeqMonk (*Chrysanthou et al., 2018*; *Latos et al., 2015*). We found that 46 of DEGs were expressed in both cell types, and of these, 26 were highly enriched in the TS compared to the ES cells (*Figure 7—figure supplement 2*).

Functional analysis revealed that the DEGs enriched in TS cells have been implicated in the control of extracellular exosome, whereas those enriched in ES cells are involved in regulating gene expression and extracellular space (*Figure 7—source data 1*). These data suggested that p110α may modulate different types of genes in the trophoblast and fetal cell lineages of the conceptus and that some, but not all of the gene changes identified in our Hom-P *versus* Het-U analysis may be a direct consequence of p110α loss in the trophoblast.

To explore these notions further, we deleted exons 18–19 of the *Pik3ca* gene using CRISPR/Cas9, to diminish p110α protein expression in TS cells (*Figure 7A–C*). We then quantified the expression of a subset of DEGs identified in vivo in the mutant p110α and wild-type TS cells (*Figure 7D*). These experiments were additionally performed in ES cells to assess if loss of p110α would have a similar effect on gene expression, as observed in mutant TS cells. Loss of p110α significantly reduced the expression of *B2m* in mutant TS cells (*Figure 7D*), similar to findings in vivo from the Hom-P *versus* Het-U placenta (*Figure 6—source data 1*). We also observed a trend for reduced expression of *Mzp12* upon TS p110α loss, although this did not reach statistical significance with the small number of biological replicates used (p=0.079; *Figure 7D*). However, none of the other DEGs analyzed were affected by TS p110α deficiency. These data suggest that the mis-regulated pattern of genes identified in the Hom-P *versus* Het-U placenta represent a combination of both direct and secondary effects of trophoblast p110α loss.

Loss of p110α also had a more prominent impact on the expression of DEGs in ES cells. For instance, p110α deficiency significantly increased the expression of *Cited2, Pdlim3, Creb3l1, Mgst1, Aldh1a3* in the ES cells (*Figure 7D*). For wildtype clones, the expression of the *Pik3ca* was overall, more uniform in ES than TS cells (*Figure 7B*). Moreover, loss of p110α tended to increase the expression of genes analyzed in ES cells, but decreased them in TS cells. Taken together, these data demonstrate that pathways downstream of p110α differentially affect the expression of genes in the trophoblast and fetal cell lineages of the conceptus.

## Discussion

Here, using in vivo conditional genetic manipulations of p110α we have identified how the trophoblast and fetal cell lineages interact to control placental resource supply and thereby, affect healthy growth of the fetus. The major cause of intrauterine growth restriction is placental insufficiency in the developed world (*Baschat and Hecher, 2004*). Our results are therefore, important for understanding the etiology of placental insufficiency and intrauterine growth restriction.

The data presented here indicate that p110α in the fetal, but not the trophoblast compartment of the conceptus determines placental size in late mouse pregnancy. The mechanism by which loss of fetal p110α reduces formation of the placenta is unknown. Elongation of the fetal capillary network is required for growth of the placenta, particularly its labyrinthine zone (*Adamson et al., 2002*; *Coan et al., 2004*). Reduced fetal capillary formation in the labyrinthine zone of the Het-F could

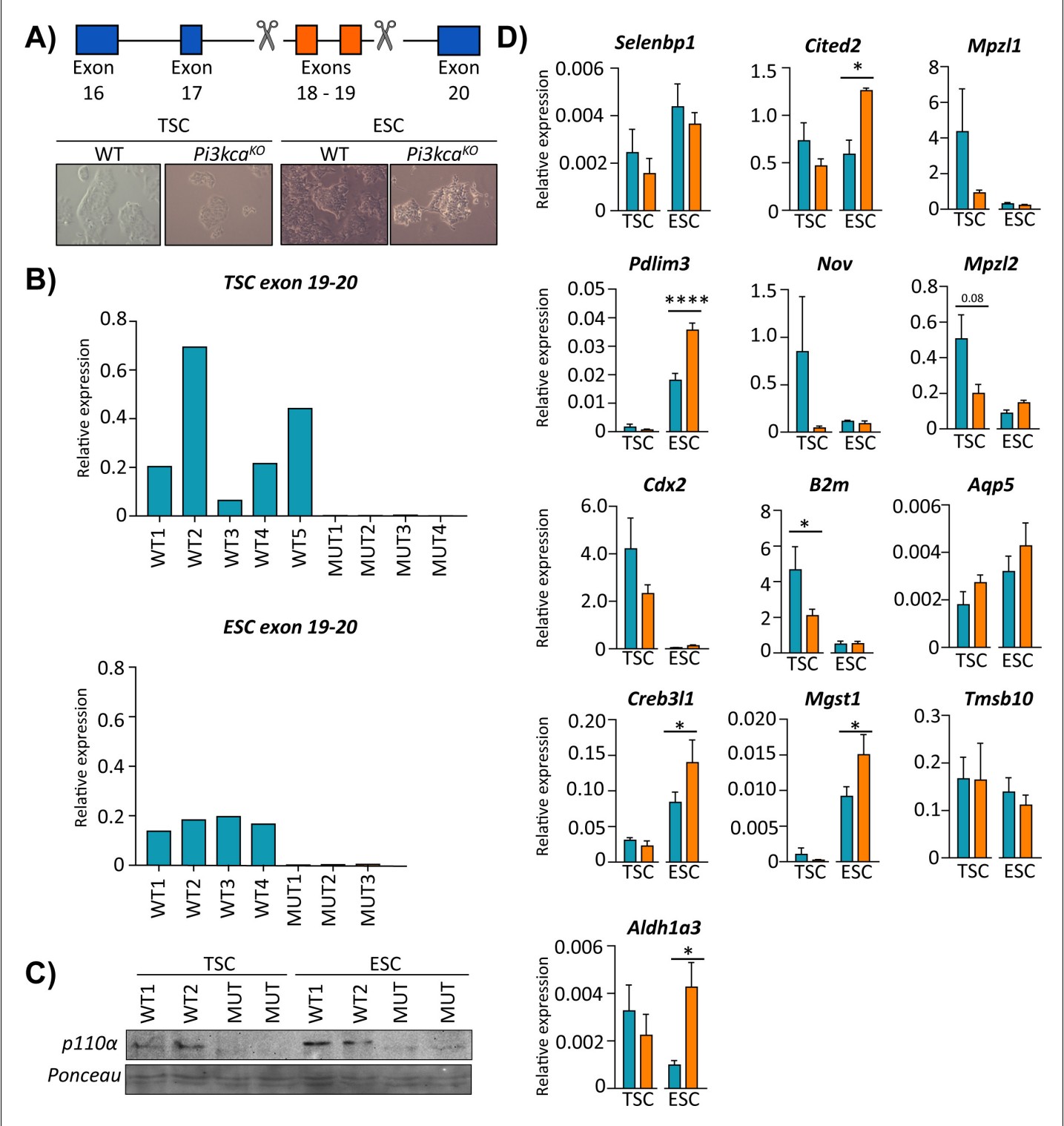

**Figure 7.** PI3K p110α differentially affects the expression of genes in the trophoblast and fetal cell lineages. (A) Details of CRISPR/Cas9 design to delete exons 18–19 of *Pik3ca* in trophoblast or embryonic stem cells (TSC and ESC, respectively). The deletion in TSC (trophoblast stem cells) and ESC (embryonic stem cells) was confirmed by (B) qRT-PCR analysis of exons 18–19 of *Pik3ca* and (C) by Western blotting for p110α. (D) The expression of candidate genes in *Pi3kca* wildtype (WT) and mutant (Mut) TSC (n = 5 and n = 4, respectively) and ESC (n = 4 and n = 3, respectively). Data in (D) are normalized against housekeeping *Sdha*. *p<0.05, ****p<0.0001, two-ways ANOVA analysis followed by Fisher post hoc test. Data presented as means ± SEM.

DOI: https://doi.org/10.7554/eLife.45282.022

*Figure 7 continued*

The following source data and figure supplements are available for figure 7:

**Source data 1.** Functional predictions for the candidate genes enriched in trophoblast and embryonic stem cells cells (TSC and ESC, respectively).
DOI: https://doi.org/10.7554/eLife.45282.025

**Source data 2.** Primer sequence for RT-qPCR, PCR screening and CRISPR gRNAs target.
DOI: https://doi.org/10.7554/eLife.45282.026

**Figure supplement 1.** Localisation and abundance of CITED2, ACTA2 and LUM identified to be dysregulated at the gene level, in Hom-P *versus* Het-U placentas on day 19 of pregnancy.
DOI: https://doi.org/10.7554/eLife.45282.023

**Figure supplement 2.** Expression profiles of the genes identified as dysregulated in Hom-P versus Het-U placentas in trophoblast stem cells (TS cells) and embryonic stem cells (ES cells), as determined by analysis of existing RNA-seq datasets (*Chrysanthou et al., 2018*; *Latos et al., 2015*).
DOI: https://doi.org/10.7554/eLife.45282.024

therefore underlie the reduced placental weight and labyrinthine zone observed and data are consistent with previous work showing defective angiogenesis and vascular development in the fetus and placenta in response to p110α loss (*Graupera et al., 2008*; *Sferruzzi-Perri et al., 2016*). The reduction in fetal and placental weight was similar for Het-F conceptuses (both reduced by ~11%), suggesting that perhaps the placenta grew to match the genetically-determined growth requirements of the fetus (*Sandovici et al., 2012*). However, it is also plausible that placental growth decreased to match the experimentally induced slower growth of the Het-F fetuses. Herein, the direct effects of fetal p110α cannot be isolated from the indirect effects on the fetal growth restriction in Het-F on day 19 of pregnancy. Thus, future work should assess the ontogeny of changes in placental phenotype with respect to the growth of the fetus in the Het-F during gestation.

The lack of an effect of trophoblast p110α deficiency in Het-P and Hom-P on placental weight was surprising as both loss of IGF2 and protein kinase B (AKT), which are main up- and down-stream mediators of PI3K signaling respectively cause placental growth restriction (*Constância et al., 2002*; *Kent et al., 2012*; *Plaks et al., 2011*; *Yang et al., 2003*). Moreover, studies using non-isoform specific inhibitors have shown that Class 1A PI3Ks mediate the proliferative and survival effects of ligands on trophoblast in vitro (*Diaz et al., 2007*; *Forbes et al., 2008*). Deleting p110α using the conditional *Pik3ca* line in the current study does not alter the stoichiometry of PI3K complexes (*Graupera et al., 2008*), and thus it is unlikely that other PI3K isoforms compensate for the loss of p110α in the trophoblast. In the current study however, the expression of cell death genes were down-regulated and the level of apoptosis in the Hom-P compared with Het-U placenta paradoxically upregulated on day 19 of pregnancy (~1 day prior to term). Differences between gene expression and tissue levels of apoptosis in our study may be relate to the low volume density (~20%) of the junctional zone in the whole placenta which was used in the RNA analysis. Assessing the ontogeny of changes in cell death genes and the activation of apoptosis in the Hom-P placenta is required. Moreover, whether placental weight (as well as morphology) may be decreased in Hom-P *versus* Het-U in the last 24 hours prior to delivery remains to be determined.

Our stereological analyses of the Het-F, Het-P and Het-U placentas demonstrated that epiblast-derived and trophoblast p110α have non-overlapping contributions in regulating the development of the placental exchange region. In particular, p110α in the epiblast-derived compartment determined the size of the placental labyrinthine zone and its trophoblast constituent, whereas p110α in the trophoblast was important for expanding the surface area for maternal-fetal exchange. Moreover, p110α in both the trophoblast and epiblast-derived lineages contributed to the formation of the capillary network and the thinness of the diffusion barrier in the placental exchange region. These findings are consistent with previous work demonstrating that the PI3K pathway regulates trophoblast differentiation (*Kent et al., 2010*; *Lee et al., 2019*) and defects in p110α signaling impair developmental angiogenesis and placental exchange region morphogenesis (*Graupera et al., 2008*; *Lelievre et al., 2005*; *Sferruzzi-Perri et al., 2016*). However, we did not find expression of any markers of known differentiated trophoblast subtypes (eg *Syna, Synb, Gcm1, Ctsq*), endothelial cells (eg *Pecam1*) or angiogenesis (eg *Ang1/2, Tie2, Vegf*) in placentas that lacked p110α (in the Hom-P compared to Het-U placenta by RNA-seq) thatwould explain the morphological phenotype observed. As fetal capillary volume was affected by a deficiency of p110α signaling in the trophoblast and trophoblast volume was altered by a deficiency in fetal p110α, our data suggest that there

is a paracrine communication between cellular compartments in the placental labyrinthine zone, which occurs via p110α signaling. Paracrine signaling via p110α may also explain why when comparing the three heterozygote lines, the Het-P placenta uniquely showed less maternal blood spaces and a greater trophoblast volume. Paracrine communication between trophoblast and endothelial cell lineages in the labyrinthine zone has been reported previously for other signaling proteins (*Limbourg et al., 2005*; *Lu et al., 2013*; *Moreau et al., 2014*; *Tang et al., 2006*). Future work should evaluate the density of different trophoblast cell types in the labyrinthine zone (eg. syncytial trophoblast layer 1, syncytial trophoblast layer two and sinusoidal giant cells) in the Het-P and other p110α mutant placentas.

Our analysis of placental transfer function in vivo revealed that independent of specific changes in placental labyrinthine morphology with fetal *versus* trophoblast p110α loss, fetal glucose and System A-mediated amino acid supply relative to the size of the fetus, was maintained or even increased due to an adapting placenta which increased materno-fetal substrate clearance per unit surface area. We found that in conceptuses with p110α deficiency, this adaptive up-regulation of placental amino acid transport depended on the expression of p110α in the trophoblast; System A-mediated amino acid transport was significantly lower for the Hom-P compared to the adapted Het-U placenta. This inability of the Hom-P placenta to up-regulate System A amino acid transport, may explain, at least partly, the further reduction in fetal weight, when compared to Het-U. Indeed, inhibiting System A-mediated amino acid transport during rat pregnancy in vivo leads to fetal growth restriction (*Cramer et al., 2002*) and reductions in System A transporter activity occur prior to the onset of intrauterine growth restriction in rats and baboons (*Jansson et al., 2006*; *Pantham et al., 2016*). Previous work has shown that constitutive heterozygous loss of p110α activity in the conceptus (α/+mutants) does not affect the expression of glucose (*Slc2a*) or the sodium coupled System A amino acid transporter (*Slc38a*) genes in the placenta (*Sferruzzi-Perri et al., 2016*). We also did not find differential expression of these transporters in Hom-P compared to Het-U placenta by RNA-seq. In other cell types, p110α alters the activity of glucose and sodium transporters on the plasma membrane (*Frevert and Kahn, 1997*; *Katagiri et al., 1996*; *Wang et al., 2008*). Loss of p110α may therefore, alter glucose and amino acid transport to the fetus by the placenta *via* post-transcriptional mechanisms. Previous work has shown that when placental supply and fetal demand for resources to grow are mis-matched, IGF2 in the fetus may directly or indirectly signal demand to the placenta to adaptively up-regulate its transport of glucose and amino acids (*Angiolini et al., 2011*; *Constância et al., 2005*; *Sferruzzi-Perri et al., 2017*; *Sferruzzi-Perri et al., 2011*). Our data therefore imply that demand signals of the compromised feto-placental unit, such as IGF2, operate via p110α in the trophoblast to adapt amino acid supply to the fetus for growth. However, it is equally possible that fetal demand signals operate via pathway/s independent of p110α. Moreover, it is plausible that these pathway/s may be sufficient to compensate for the effects of moderately down-regulated p110α in the Het-P, but are insufficient when p110α is markedly knocked-down as in the Hom-P. Further work is required to identify the elusive fetal demand/s signals and explore the contribution of other signaling pathways implicated in environmental sensing, in driving alterations in placental transport phenotype in response to p110α deficiency, including the mechanistic target of rapamycin (mTOR), mitogen-activated pathway (MAPK), general control nonrepressed 2 (GCN2), glucokinase, G-protein coupled-receptors (GPCRs) and adenosine monophosphate-activated protein kinase (AMPK) (*Efeyan et al., 2015*).

Our transcriptomic comparison of the Hom-P and Het-U placentas identified several genes downstream of p110α that may be important in adapting placental phenotype to support fetal growth. Of note, we found six granzyme encoding genes, were down-regulated in Hom-P compared with Het-U. Others have shown that the PI3K pathway regulates granzyme expression and function of different leukocyte populations (*Blanco et al., 2015*; *Efimova and Kelley, 2009*; *Jiang et al., 2000*), and thus p110α could be playing a similar role in trophoblast. We also found that some of the genes differentially expressed between Hom-P and Het-U placenta have been previously implicated in the regulation of trophoblast differentiation (*Cdx2*) (*Strumpf et al., 2005*), growth and transport (*Cited2*)(*Withington et al., 2006*) and hormonal regulation of maternal physiology (*Prl5a1*, *Prl7c1*, *Psg19* and *Psg22*) (*Moore and Dveksler, 2014*; *Soares et al., 2007*). Moreover, a number of genes identified have been implicated in miscarriage (*Fgl2*) (*Clark et al., 2001*) and in the development of placental dysfunction in the pregnancy complication, preeclampsia (*Acta2*, *Ngf* and *Nov/Ccn3*) (*Gellhaus et al., 2006*; *Sahay et al., 2015*; *Todros et al., 2007*). We found significant enrichment of

binding motifs for the transcription factors, STAT1, ARNT2, HMGA1 and MITF in the promoters of genes differentially expressed between Hom-P and Het-U. Others have demonstrated that the PI3K pathway modulates the activity of three of these transcription factors (*Mounayar et al., 2015*; *Sheu et al., 2017*; *Terragni et al., 2011*; *Zhang et al., 2018*) and of note, there is an interaction between p110α and STAT1 activity (*Mounayar et al., 2015*). Many of the differentially expressed genes in Hom-P *versus* Het-U placenta, were found to be enriched in TS cells however, of the fourteen candidates studied, only one was significantly altered in response to TS cell p110α deficiency. Moreover, several of the genes/proteins differentially expressed in Hom-P *versus* Het-U placenta (*Cited2*/CITED2, *Acta2*/ACTA2, *Lum*/LUM) were localized to both trophoblast in junctional and labyrinthine zones and fetal constituents including labyrinthine fetal vessels in the placenta. Thus, the mis-regulated pattern of genes identified in the Hom-P *versus* Het-U placenta may be both a direct consequence of p110α in the trophoblast, as well as the result of indirect effects on the embryonic lineages (such as fetal capillaries) in the placenta, or in the decidua. Single-cell RNA-seq should be performed on p110α mutant placentas in the future to substantiate this interpretation. Further work is additionally required to assess more directly the contribution of genes differentially expressed between Hom-P and Het-U placentas, in adaptive responses of the placenta (*Sferruzzi-Perri et al., 2011*).

Loss of p110α in ES cells also affected the expression the candidate genes studied. However, there were many more candidate genes significantly altered in the mutant ES cells and the pattern of change was largely distinct to that observed for the mutant TS cells. Indeed, genes were typically upregulated in ES cells, but reduced in TS cells in response to p110α deficiency. Moreover, the genes differentially expressed between Hom-P and Het-U and enriched in TS and ES cells were predicted to participate in diverse cell functions, for example extracellular exosome and gene expression, respectively. These data imply that p110α regulates the expression of different genes and functional pathways in the trophoblast and epiblast-derived tissues of the conceptus. They also reinforce that the embryonic lineages of the conceptus are more sensitive to p110α loss than the trophoblast and may provide some explanation for the lethality observed for the Hom-F but not Hom-P mutants. These findings could be furthered in the future by performing RNA on TS and ES cells to obtain a complete transcriptomic comparison in response to p110α deficiency. However, it is important to note that TS and ES cells would not have been present in the wildtype or p110α mutant placentas at the time of investigation (day 19 of pregnancy). Thus, extrapolating our findings of p110α deficiency in ES and TS cells in vitro to our trophoblast or epiblast-derived manipulations in vivo, are limited and should be taken cautiously.

Previous work has shown that the genotype and environment of the mother modulates placental growth and transport function (*Angiolini et al., 2011*; *Constância et al., 2005*; *Sferruzzi-Perri et al., 2016*; *Sferruzzi-Perri et al., 2017*; *Sferruzzi-Perri et al., 2011*). Moreover, litter composition and particularly, the presence of siblings of a different genotype in the litter can affect placental phenotype (*Tunster et al., 2016*). Indeed, in the current study, some of the values for the placenta of the Het-F wildtype controls were higher than for the Het-U and Het-P (eg placental weight, volume of labyrinthine zone trophoblast and fetal capillaries and surface area), which likely reflects variations in the gestational environment (litter composition and maternal genotype/environment) for the individual groups. All data in the current study were obtained from litters of mixed genotype and only comparisons to their respective control siblings were made to minimize possible effects of variations in the gestational metabolic environment on placental phenotype. However, it is important to note that changes in placental phenotype with fetal and trophoblast *Pik3ca* deletion can still occur within the homogenous gestational environment and according to fetal genotype, which was not assessed in the current study. Future work should, therefore, attempt to explore the interactions between maternal p110α metabolic profile, litter composition and fetal genotype in the regulation of placental resource allocation. This can, for instance, be done initially by measuring maternal hormones and glucose/amino-acid levels in the diverse fetal/maternal genotype combinations, but quantification of the 'individual' effects will be highly challenging.

Collectively, our data indicate that p110α plays distinct roles in the different compartments of the conceptus, which ultimately affect placental phenotype and fetal growth. In particular, p110α in the fetal lineages of the conceptus is essential for prenatal development and a major regulator of placental phenotype (*Figure 8*). Moreover, trophoblast p110α signaling is critical for its ability to upregulate amino acid transport to match fetal demands for growth near term. However, the Hom-F

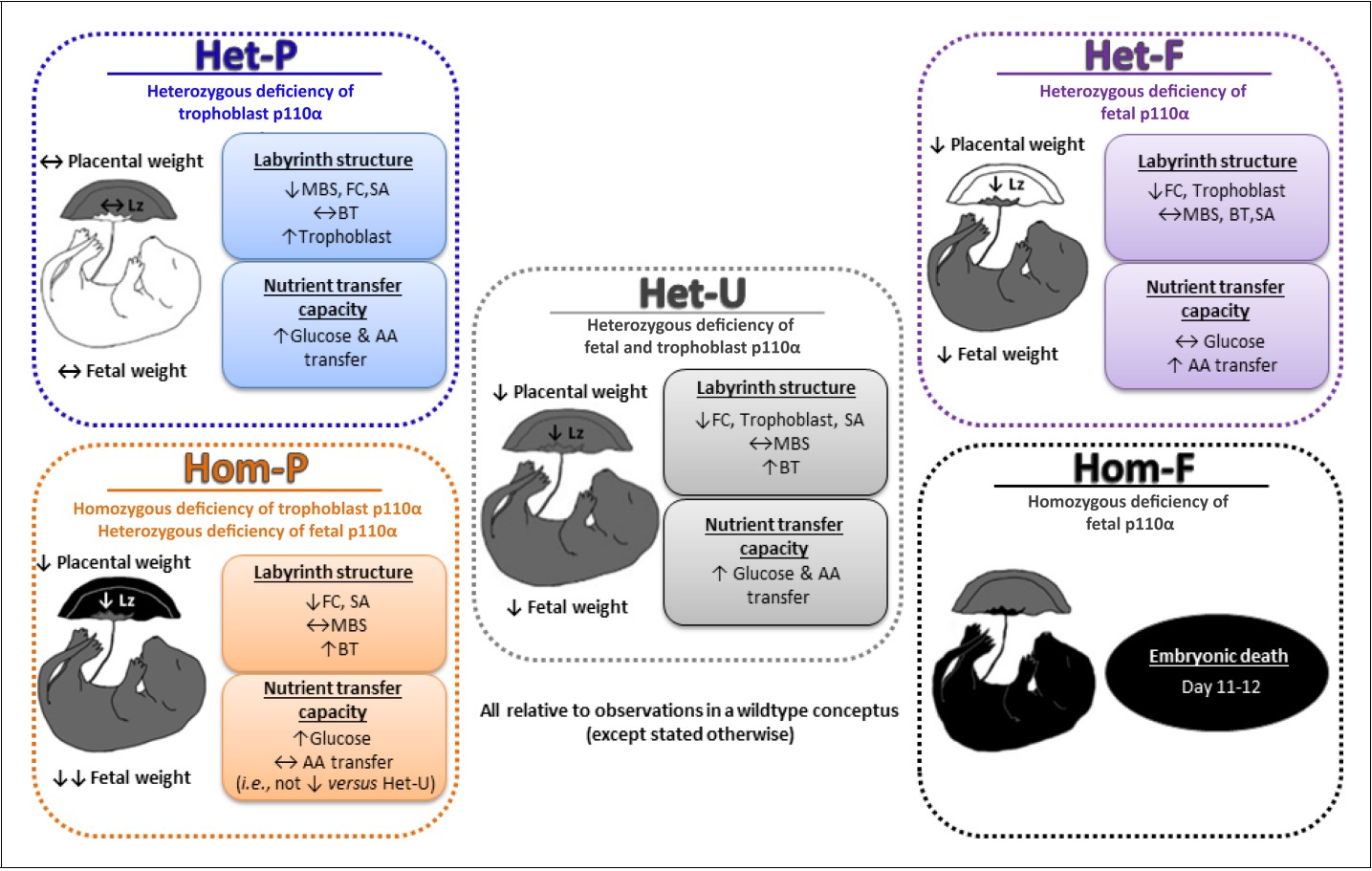

**Figure 8.** Summary figure showing the phenotypes of all p110α mutants used in this study, which together highlight that fetal and trophoblast p110α have distinct contributions in regulating resource allocation to the growing fetus. Findings are from mice at day 19 of pregnancy, unless indicated otherwise. AA=amino acid, BT=barrier thickness, FC=fetal capillaries, Lz=labyrinth zone, MBS=maternal blood spaces, SA=surface area for exchange.
DOI: https://doi.org/10.7554/eLife.45282.027

and Hom-P have a heterozygous deficiency of p110α in the trophoblast and epiblast-derived line-ages, respectively. Thus, there may be interactions between the different compartments of the con-ceptus that are heterozygous and homozygous deficient in p110α, which may have contributed to the placental and fetal phenotypes observed for Hom-F and Hom-P. Furthermore, gene disruption may occur up to 1.5 days earlier using the *Meox2*Cre, compared to the *Cyp19*Cre (at around days 5 and 6.5, respectively) (*Tallquist and Soriano, 2000*; *Wenzel and Leone, 2007*). Moreover, the *Cyp19*Cre exhibits mosaic activity (observed herein and reported previously; *Wenzel and Leone, 2007*). Therefore, temporal differences in the timing of Cre activation and levels of Cre activity between the *Meox2*Cre and *Cyp19*Cre lines may have contributed to feto-placental phenotypes observed for the fetal *versus* trophoblast p110α mutants, in the current study. In addition, previous characterization of *Meox2*Cre line has shown Cre is active in the yolk sac mesoderm (*Tallquist and Soriano, 2000*). As the yolk sac is essential for fetal growth, loss of p110α from the yolk sac meso-derm could have contributed to the fetal phenotypes observed with the Het-F and Hom-F. Employ-ing alternative strategies to target expression of p110α in different conceptus lineages, such as using lentiviral-mediated shRNA/siRNA transfection of mouse blastocysts (*Georgiades et al., 2007*; *Okada et al., 2007*) are therefore, warranted in the future. Nevertheless our findings are important in the context of human pregnancy, as dysregulated expression of both up and down-stream com-ponents of the PI3K pathway in the placenta are associated with abnormal fetal growth and poor pregnancy outcome (reviewed in *Sferruzzi-Perri et al., 2017*).

## Materials and methods

### Mice and genotyping

Mice were housed under dark:light 12:12 conditions with free access to water and chow in the University of Cambridge Animal Facility abiding by the UK Home Office Animals (Scientific Procedures) Act 1986 and local ethics committee. Home Office project license number is 70/7645. Mice in which exons 18 and 19 of the p110α gene were flanked by LoxP sites (*Pik3ca^Fl*) on a C57Bl6 background were kindly provided by Dr Bart Vanhaesebroeck (University College London) and Dr Klaus Okkenhaug (Babraham Institute) (*Graupera et al., 2008*). These were time-mated with those expressing the Cre recombinase gene under the control of the trophoblast-specific human *Cyp19* promoter (*Wenzel and Leone, 2007*), the embryonic *Meox2* promoter (*Tallquist and Soriano, 2000*) or the ubiquitous *CMV* promoter (*Schwenk et al., 1995*). This achieved three models that had a ~ 50% decrease in p110α in the trophoblast, fetal lineages or whole conceptus, termed Het-P, Het-F and Het-U, respectively. Further crosses were performed on a second generation of Het-U mice to selectively delete the remaining p110α allele in the trophoblast using *Cyp19*Cre (Hom-P) or fetal lineage using *Meox2*Cre (Hom-F). See *Figure 1—source data 1* for experimental crosses used. Note the wild-types for the Het-P and Het-U are the same animals and the Het-U were the controls for Hom-P. The genotypes of the fetuses/mice were determined by PCR using primers to detect *Cyp19*Cre (Fwd: GACCTTGCTGAGATTAGATC and Rev: AGAGAGAAGCATGTTTAGCTGG), *Meox2*Cre (Fwd: GGACCACCTTCTTTTGGCTTC, Rev: AAGATGTGGAGAGTACGGGGTAG, Cre: CAGATCCTCCTCAGAAATCAGC) and *CMV*Cre (Cre Fwd: CGAGTGATGAGGTTCGCAAG, Cre Rev: TGAGTGAACGAACCTGGTCG, Internal control Fwd: ATGTCTCCAATCCTTGAACACTG and Internal control Rev: GCAGTGGGAGAAATCAGAACC). This was done using the following cycle conditions: 94℃, 3 min, then 35 cycles of 94℃ for 30 s, 57℃ for 30 s and 72℃ for 90 s, and then 72℃ for 7 min. Genotyping was also performed by PCR using primers for *Pik3ca^Fl* (Fwd:CTAAGCCCCTTAAAGCCTTAC, Rev: CAGCTCCCATCTCAGTTCA and Deletion: ACACACTGCATCAATGGC) as described previously (*Graupera et al., 2008*) and by real time PCR (qRT-PCR) using Taqman probes (*Pik3ca*: Mm00435672_g1 and *Gapdh*: Mm99999915_g1) to confirm deletion in whole fetal and placental tissue homogenates. Due to the mosaic activity of the *Cyp19*Cre (*Wenzel and Leone, 2007*) a cut-off for *Pik3ca* deletion in the placenta using qRT-PCR was set for <65% Het-P and <30% for Hom-P.

To visualize the Cre recombinase activity, *Cyp19*Cre and *Meox2*Cre mice were mated to the *Rosa26*TdRFP reporter mouse line (*Gt(ROSA)26Sor;* purchased from the Jackson lab and a generous gift from Prof William Colledge, University of Cambridge) (*Soriano, 1999*). Fetuses and placentas were collected on day 16 of pregnancy, fixed in 4% (wt/vol) PFA, 30% (wt/vol) sucrose and then frozen in OCT (optimal cutting temperature media, TissueTek) for cryosectioning and staining in DAPI.

### Placental nutrient transfer assays

The unidirectional materno-fetal clearance of the non-metabolisable radioactive tracers, $^{14}$C-methyl amino-isobutyric acid (MeAIB) and $^{3}$H-methyl D-Glucose (MeG) were measured under anesthesia with fentanyl-fluanisone (hypnorm):midazolam (hypnovel) in sterile water (1:1:2, Jansen Animal Health) (*Sibley et al., 2004*). A 200 µl bolus containing 3.5 µCi, of MeAIB (NEN NEC-671; specific activity 1.86GBq/mmol, Perkin Elmer, USA) and 3.5 µCi MeG (NEN NEC-377; specific activity 2.1GBq/mmol) in physiological saline (0.9% wt/vol) was injected into the maternal jugular vein. Two minutes after tracer injection, the dam was killed by cervical dislocation, uteri were collected and fetal and placental weights recorded. Entire litters of placentas were collected for morphological analyses or snap frozen in liquid nitrogen for quantification of gene expression. Fetuses were decapitated, entire fetal tails taken for DNA genotyping and then fetuses minced and lysed at 55℃ in Biosol (National Diagnostics, Atlanta, USA). Fetal lysates were measured for beta emissions by liquid scintillation counting (Optiphase Hisafe II and Packard Tri-Carb, 1900; Perkin-Elmer USA) and radioactivity (DPM; disintegrations per minute) in the fetuses was used to calculate transfer relative to the estimated placental surface area or to fetal weight.

### Placental morphological analyses

Placentas were bisected and one half was fixed in 4% (wt/vol) paraformaldehyde, paraffin-embedding, exhaustively sectioned at 7 µm and stained with hematoxylin and eosin to analyse gross

placental structure. The other half was fixed in 4% (wt/vol) glutaraldehyde, embedded in Spurr's epoxy resin and a single, and 1 μm midline section was cut and then stained with toluidine blue for detailed analysis of labyrinthine zone (Lz) structure using the Computer Assisted Stereological Toolbox (CAST v2.0) program as previously described (Coan et al., 2004). Briefly, to determine the volume densities of each Lz component (fetal capillaries [FC], maternal blood spaces [MBS] and trophoblast), point counting was used and their densities were multiplied by the estimated volume of the Lz to obtain estimated component volumes. Surface densities of maternal-facing and fetal-facing interhaemal membrane surfaces were then determined by counting the number of intersection points along cycloid arcs in random fields of view. These were converted to absolute surface areas and the total surface area for exchange calculated (averaged surface area of MBS and FC). The mean interhaemal membrane thickness was determined by measuring the shortest distance between FC and the closest MBS at random starting locations within the Lz. Per Lz, 200 measurements were made. The harmonic mean thickness (barrier thickness) was calculated from the reciprocal of the mean of the reciprocal distances.

## Placental in situ cell death staining

Cells undergoing cell death were detected in paraffin-embedded sections of placenta using the In Situ Cell Death Detection Kit, TMR red (Sigma Aldrich), according to the manufacturer's instructions. The proportion of TMR red to DAPI stained cells was determined in each placental section using Image J (freeware software).

## Placental immuno-localisation of protein abundance

For localization of p110α, placental sections were washed with PBS to remove OCT and underwent antigen retrieval with citrate buffer before immunolabelling (antibody against p110α, Cell Signaling, C73F8, 1:75 dilution). Sections were treated with 0.5% Triton X-100 before immunolabelling. Bound antibody was detected using biotinylated goat anti-rabbit IgG (Abcam, ab6720) followed by streptavidin-conjugated horseradish peroxidase (Rockland, S000-03) and 3,3-diaminobenzidine (DAB) according to manufacturer instructions. Sections were lightly counterstained with hematoxylin and mounted in DPX. Negative controls were performed by omission of the primary antibody. For ACTA2 (R and D, MAB1420-SP, 5 μg/ml dilution), LUM (R and D, AF2745-SP, 5 μg/ml dilution), CITED2 (R and D, AF5005-SP, 3 μg/ml dilution) and Dolichos Biflorus Agglutinin lectin (DBA; Vector; B-1035; 6.6 μg/ml dilution) placental sections were de-paraffinized, rehydrated and subject to antigen retrieval prior to immunolabelling. Bound antibody against DBA, CITED2, ACTA2 and LUM was detected using biotinylated goat anti-rabbit (Abcam, ab6720), donkey anti-sheep (Abcam, ab97125), horse anti-mouse (Vector, BA-2000) and horse anti-goat (Vector, BA-9500), respectively. This was followed by incubation with streptavidin-conjugated horseradish peroxidase and visualization with DAB staining, as described previously. Sections were lightly counterstained with nuclear fast red and mounted in DPX. Analysis of each placental section was performed using Image J (freeware software).

## Placental RNA sequencing (RNA-seq)

Total RNA was extracted from whole placental halves using the RNeasy Plus Mini Kit (Qiagen, UK) according to manufacturer's instructions. RNA-seq libraries were prepared using TruSeq Stranded total RNA library preparation kit (Illumina) and barcoded libraries were combined and submitted for sequencing (50 bp single ended on a HiSeq2500) by the Wellcome Trust-MRC Institute of Metabolic Science Genomics Core Facility. The number of raw reads per library was between 14.3 and 33.4 million. Fastq files were aligned to mouse genome GRCm38 with TopHat (version 2.0.11; Kim 2013) using default settings (The mapping percentage varied between 96.2% and 98.3%). Differential expression was performed using Cuffdiff (version 2.1.1; Trapnell 2012) with the following settings; require a minimum of 5 alignments to conduct significance testing, perform bias correction, do initial estimation to weight multi-hits. Genes identified as differentially expressed (DEG) with Cuffdiff significance indicated as q value < 0.05 and 1.5-linear fold expression change. Pathway analysis was then determined using DAVID gene ontology (Huang et al., 2009) and redundant GO terms were filtered-out using REViGO (Supek et al., 2011). The RNA-seq data have been deposited in NCBI's

Gene Expression Omnibus and are accessible through GEO Series accession number GSE126046 (https://www.ncbi.nlm.nih.gov/geo/query/acc.cgi?acc=GSE126046).

To search for enrichment of transcription factor binding sites at the promoters of DEG, we used Eukaryotic Promoter Database (EPD – https://epd.vital-it.ch/index.php) to retrieve the DNA sequences from 1,000 bp upstream to 100 bp downstream of the transcriptional start site. These sequences were then analysed using Analysis of Motif Enrichment (AME v4.12.0 – http://meme-suite.org/tools/ame) by selecting *Mus musculus* and HOCOMOCO Mouse (v11 FULL) as motif database. Transcription factors predicted by AME and Ingenuity Pathway Analysis (IPA) were then used for network visualization, performed using IPA.

## CRISPR/Cas9 knockout in Trophoblast and Embryonic Stem Cells

For CRISPR/Cas9-mediated deletion of *Pik3ca*, *E14tg2a* embryonic stem cells (obtained from the Mutant Mouse Regional Resource Center, UC Davis, USA) and *TS-Rs26* trophoblast stem cells (a gift of the Rossant lab, Toronto, Canada) (both cell lines were tested and negative for mycoplasma) were transfected with Cas9.2A-eGFP plasmid (Plasmid 48138 Addgene) harboring guide RNAs flanking the exon 18 and exon 19 (http://CRISPR.mit.edu) (*Figure 7*). Transfection was carried out with Lipofectamine 2000 (ThermoFisher Scientific 11668019) reagent according to the manufacturer's protocol. Knockout clones were confirmed by western blotting and by genotyping using primers spanning the deleted exons, and by reverse transcription followed by semi-quantitative PCR (RT-qPCR) with primers (AGGGAGCACAAGAGTACACCA and GGCATGCTGCCGAATTGCTA) within and downstream of the deleted exons, as shown in *Figure 7*. Values were normalized to *Sdha* expression. Three to five independent knockout clones were analysed for each cell line. The expression of fourteen genes found to be differentially regulated between Het-U and Hom-P placenta, were then assessed by qPCR in p110α deficient TS and ES cells.

## Gene expression by qRT-PCR

Total RNA was extracted from whole placental halves using the RNeasy Plus Mini Kit (Qiagen, UK) whereas total RNA was extracted from TS and ES cells using trizol (Invitrogen), chloroform and ethanol extraction. Multiscribe Reverse Transcriptase and random primers (Applied Biosystems) were used to synthesize cDNA from 2.5 ug of RNA. Samples were analysed in duplicate by qRT-PCR (7500 Fast Real-Time PCR System, Applied Biosystems, UK) using Sybr green chemistry and pairs of forward and reverse primers (*Figure 7—source data 2*). The expression of genes of interest were normalized to the expression of *Gapdh* or *Sdha* which was not affected by genotype. Data were analysed using the 2–ΔΔCT method for quantification.

## Western blotting

Protein expression was quantified using p110α antibody (1:500; Cell Signaling, 4249), as described previously (*Sferruzzi-Perri et al., 2011*).

## Statistics

No explicit power analyses were used to predetermine sample size. Randomization was not used in our studies. Analyses of placental morphology and in situ cell death staining were conducted blinded to the genotype. The number of samples per group for each analysis is detailed in the legends of figures and footnotes of tables. Data are presented as mean ± SEM. Data were considered normally distributed and analysed using unpaired and paired t tests with Excel, as required or two-way ANOVA with GraphPad Prism 7. Data were considered statistically significant when $p < 0.05$.

## Acknowledgements

We thank the Centre for Trophoblast Research for the award of a Next Generation Fellowship and the Royal Society for a Dorothy Hodgkin Research Fellowship to ANS-P. We also thank the Royal Society for a Newton International Fellowship, the Erasmus Exchange scheme and COST ACTION (EPICONCEPT and SALAAM) for grants to J L-T. The RNA-seq work was performed with the genomics and transcriptomics core, which is funded by the UK Medical Research Council (MRC) Metabolic Disease Unit (MRC_MC_UU_12012/5) and a Wellcome Trust Major Award (208363/Z/17/Z). We

thank Professor Abigail Fowden for allowing us to perform the animal procedures under her project license and the staff at the Combined Animal Facility for their technical help. We thank Dr Bart Vanhaesebroeck and Dr Klaus Okkenhaug for providing the p110α-floxed mice and Professor William Colledge for the *tdTomato* mice. Finally, we thank Dr Jacqui Shields and Dr Angela Riedel for scanning our fluorescently labeled slides and technical support.

## Additional information

### Funding

| Funder | Grant reference number | Author |
| --- | --- | --- |
| Centre for Trophoblast Research | Next Generation Fellowship | Amanda N Sferruzzi-Perri |
| Royal Society | Dorothy Hodgkin Fellowship | Amanda N Sferruzzi-Perri |
| Royal Society | Newton International Fellowship | Jorge López-Tello |
| European Cooperation in Science and Technology | SALAAM | Jorge López-Tello |
| Erasmus+ | | Jorge López-Tello |
| European Cooperation in Science and Technology | EPICONCEPT | Jorge López-Tello |

The funders had no role in study design, data collection and interpretation, or the decision to submit the work for publication.

### Author contributions

Jorge López-Tello, Data curation, Formal analysis, Validation, Investigation, Visualization, Methodology, Writing—original draft, Writing—review and editing; Vicente Pérez-García, Conceptualization, Validation, Investigation, Methodology, Writing—review and editing; Jaspreet Khaira, Laura C Kusinski, Adam Andreani, Imogen Grant, Edurne Fernández de Liger, Investigation, Methodology, Writing—review and editing; Wendy N Cooper, Data curation, Formal analysis, Investigation, Methodology, Writing—review and editing; Brian YH Lam, Software, Methodology, Writing—review and editing; Myriam Hemberger, Formal analysis, Investigation, Methodology, Writing—review and editing; Ionel Sandovici, Data curation, Software, Formal analysis, Investigation, Methodology, Writing—review and editing; Miguel Constancia, Conceptualization, Resources, Supervision, Investigation, Writing—review and editing; Amanda N Sferruzzi-Perri, Conceptualization, Resources, Data curation, Formal analysis, Supervision, Funding acquisition, Validation, Investigation, Visualization, Methodology, Writing—original draft, Project administration, Writing—review and editing

### Author ORCIDs

Jorge López-Tello (iD) https://orcid.org/0000-0002-7886-0233
Wendy N Cooper (iD) http://orcid.org/0000-0003-3416-9982
Ionel Sandovici (iD) http://orcid.org/0000-0001-5674-4269
Amanda N Sferruzzi-Perri (iD) https://orcid.org/0000-0002-4931-4233

### Ethics

Animal experimentation: All experiments were carried out in accordance with the UK Home Office Animals (Scientific Procedures) Act 1986 following ethical review by the University of Cambridge Animal Welfare and Ethical Review Board. Home Office project license number is 70/7645.

### Decision letter and Author response

Decision letter https://doi.org/10.7554/eLife.45282.036
Author response https://doi.org/10.7554/eLife.45282.037

## Additional files

### Supplementary files

• Transparent reporting form
DOI: https://doi.org/10.7554/eLife.45282.028

### Data availability

The RNA-seq data have been deposited in NCBI's Gene Expression Omnibus and are accessible through GEO Series accession number GSE126046 (https://www.ncbi.nlm.nih.gov/geo/query/acc.cgi?acc=GSE126046). All other relevant data are within the manuscript and its Supporting Information files.

The following dataset was generated:

| Author(s) | Year | Dataset title | Dataset URL | Database and Identifier |
|---|---|---|---|---|
| Lopez-Tello J, Perez-Garcia V, Khaira J, Kusinski LC, Cooper WN, Andreani A, Grant I, Fernandez de Liger E, Lam BYH, Hemberger M, Sandovici I, Constancia M, Sferruzzi-Perri AN | 2019 | Fetal and trophoblast PI3K p110α have distinct roles in regulating resource supply to the growing fetus | https://www.ncbi.nlm.nih.gov/geo/query/acc.cgi?acc=GSE126046 | NCBI Gene Expression Omnibus, GSE126046 |

The following previously published datasets were used:

| Author(s) | Year | Dataset title | Dataset URL | Database and Identifier |
|---|---|---|---|---|
| Chrysanthou S, Senner CE, Burge S, Fineberg E, Okkenhaug H, Woods L, Perez-Garcia V, Hemberger M | 2018 | A novel role of Tet1/2 proteins in cell cycle progression of trophoblast stem cells | https://www.ncbi.nlm.nih.gov/geo/query/acc.cgi?acc=GSE109545 | NCBI Gene Expression Omnibus, GSE109545 |
| Latos PA, Sienerth AR, Murray A, Senner CE, Muto M, Ikawa M, Oxley D, Burge S, Cox BJ, Hemberger M | 2015 | Elf5-centered transcription factor hub controls trophoblast stem cell self-renewal and differentiation through stoichiometry-sensitive shifts in target gene networks | https://www.ebi.ac.uk/ena/data/search?query=PRJNA298763 | European Nucleotide Archive, PRJNA298763 |

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
