## [Decision Letter]

Thank you for submitting your article "Fetal and trophoblast PI3Kp110α have distinct roles in regulating resource supply to the growing fetus" for consideration by *eLife*. Your article has been reviewed by three peer reviewers, and the evaluation has been overseen by a Reviewing Editor and Jonathan Cooper as the Senior Editor. The following individuals involved in review of your submission have agreed to reveal their identity: David Natale (Reviewer #1); Elizabeth Enninga (Reviewer #3).

The reviewers have discussed the reviews with one another and the Reviewing Editor has drafted this decision to help you prepare a revised submission.

Summary:

In this manuscript Lopez-Tello and colleagues perform a series of elegant genetic experiments to dissect the distinct functions of PI3K signalling via p110α in the portions of the conceptus derived from the epiblast and trophoblasts. They assess the effects of p110α deficiency on nutrient transfer, and fetal and placental growth. This is followed by RNA-seq analysis of gene expression in mutant placentae followed by further exploration of distinct roles of p110α function in fetal and trophoblast lineages through comparison of CRISPR knockout ES and TS cells. The data are novel and add to our understanding of fetal/placental interactions in controlling growth and development during gestation.

Essential revisions:

1) The reviewers suggest a revision of Figure 3. You present your findings regarding the size of the different placental layers, blood spaces, surface area and interhaemal thickness. The numbers provided were calculated through morphometric analysis of wildtype and mutant placentas from three different genetic models. The issue is that the wildtype controls for Het-U and Het-P samples are very similar but the Het-F wildtype controls are quite a bit higher. Why would this be the case? Furthermore, when examining the values for the p110α deficient placenta samples, they all appear to be very similar. Therefore, it would appear that the statistically significant differences that were observed in the different groups may actually be a result of the difference between the wildtype control samples. We ask that this important issue be addressed in the revised version.

2) To investigate whether genes differentially expressed between Hom-P and Het-U placentae in your bulk RNA-seq experiment are expressed in trophoblast or epiblast-derived cells, their expression is analysed in ES and TS cell datasets. However, none of the cells in the placenta at the time under investigation are ES or TS cells, and are likely to be quite different. This therefore does not seem like an ideal/appropriate comparison. A more valid and informative analysis may well be to analyse the expression patterns of the differentially expressed genes histologically in Hom-P and Het-U placentae. This would reveal where the genes are expressed and potentially if and how gene expression patterns (rather than just levels) change between Hom-P and Het-U placentae. This may permit a more informed interpretation of the RNA-seq results, which are currently somewhat confusing. Could you please indicate the limitations of this comparison in the text.

3) That granzymes and Perforin-1 are differentially expressed in the RNA-seq experiment potentially suggests an effect on uterine NK cells. Are there differences in uNK cell levels or gene expression between Hom-P and Het-U placentae? Establishing this is critical to interpretation of the data. This could easily be addressed either by performing histological analysis and staining with available reagents that readily detect NK cells. Alternatively this finding should be discussed in the text.

---

## [Author Response]

Essential revisions:1) The reviewers suggest a revision of Figure 3. You present your findings regarding the size of the different placental layers, blood spaces, surface area and interhaemal thickness. The numbers provided were calculated through morphometric analysis of wildtype and mutant placentas from three different genetic models. The issue is that the wildtype controls for Het-U and Het-P samples are very similar but the Het-F wildtype controls are quite a bit higher. Why would this be the case? Furthermore, when examining the values for the p110α deficient placenta samples, they all appear to be very similar. Therefore, it would appear that the statistically significant differences that were observed in the different groups may actually be a result of the difference between the wildtype control samples. We ask that this important issue be addressed in the revised version.

We thank the reviewers for their comments. Indeed, some of the values for the placenta of the Het-F wildtype controls are higher than for the Het-U and Het-P. However, the *Meox2*Cre, *CMV*Cre and *Cyp19*Cre lines used to generate these mutants and the corresponding wildtypes have been on a C57Bl6 background for more than 10 generations. Moreover, there are no overt differences in fetal weight between the different wildtype controls. At present we do not have a definitive explanation for the differences in WT placenta phenotype for Het-F, Het-U and Het-P mutants, other than this may be due to differences in the maternal genotype (mothers carry different Cre alleles), or litter composition (differences in the proportions of wildtypes and mutants in each litter). Indeed we and others have previously demonstrated the influence of the maternal genotype (Angiolini et al., 2011; Constancia et al., 2005; Sferruzzi-Perri et al., 2016; Sferruzzi-Perri et al., 2017; Sferruzzi-Perri et al., 2011) and sibling genotype from the same litter (Tunster et al., 2016), on the phenotype of the placenta. As a result, we intentionally only performed statistical comparisons between the mutant heterozygotes and their respective wildtype littermates. Moreover, no direct statistical comparisons of mutants from different crosses were performed. We have inserted a few sentences in the revised Discussion to address this important issue:

“Previous work has shown that the genotype and environment of the mother modulates placental growth and transport function (Angiolini et al., 2011; Constancia et al., 2005; Sferruzzi-Perri et al., 2016; Sferruzzi-Perri et al., 2017; Sferruzzi-Perri et al., 2011). […] This can, for instance, be done initially by measuring maternal hormones and glucose/amino-acid levels in the diverse fetal/maternal genotype combinations, but quantification of the ‘individual' effects will be highly challenging.”

2) To investigate whether genes differentially expressed between Hom-P and Het-U placentae in your bulk RNA-seq experiment are expressed in trophoblast or epiblast-derived cells, their expression is analysed in ES and TS cell datasets. However, none of the cells in the placenta at the time under investigation are ES or TS cells, and are likely to be quite different. This therefore does not seem like an ideal/appropriate comparison. A more valid and informative analysis may well be to analyse the expression patterns of the differentially expressed genes histologically in Hom-P and Het-U placentae. This would reveal where the genes are expressed and potentially if and how gene expression patterns (rather than just levels) change between Hom-P and Het-U placentae. This may permit a more informed interpretation of the RNA-seq results, which are currently somewhat confusing. Could you please indicate the limitations of this comparison in the text.

We thank the reviewers for their thoughtful comments. As requested, we have inserted a sentence highlighting the limitations of analysing expression of differentially expressed genes in TS and ES datasets in the revised manuscript:

“However, it is important to note that TS and ES cells would not have been present in the wildtype or p110α mutant placentas at the time of investigation (day 19 of pregnancy). Thus, extrapolating our findings of p110α deficiency in ES and TS cells in vitro to our trophoblast or epiblast-derived manipulations in vivo, are limited and should be taken cautiously.”

In addition, we have histologically assessed the expression of a few differentially expressed genes/proteins in Hom-P and Het-U placentas (*Acta2*/ACTA2, *Cited2*/CITED2, *Lum*/LUM). We found that all these proteins were localized to both trophoblast (in junctional and labyrinthine zones) and epiblast-derived compartments (including fetal vessels in the labyrinthine zone) in the placenta. However, only the abundance of CITED2 appeared to be altered in a similar direction to the its gene, in Hom-P versus Het-U placenta. We have included this new data in the revised manuscript (see Figure 7—figure supplement 1).

Results:

“Using histology, we assessed the expression of a few differentially expressed genes/proteins in the Hom-P and Het-U placenta (Acta2/ACTA2, Cited2/CITED2, Lum/LUM). […] However, only the abundance of CITED2 protein followed a similar direction to the gene, in Hom-P compared with Het-U placentas.”

Discussion:

“Moreover, several of the genes/proteins differentially expressed in Hom-P versus Het-U placenta (Cited2/CITED2, Acta2/ACTA2, Lum/LUM) were localized to both trophoblast in junctional and labyrinthine zones and fetal constituents including labyrinthine fetal vessels in the placenta.”

3) That granzymes and Perforin-1 are differentially expressed in the RNA-seq experiment potentially suggests an effect on uterine NK cells. Are there differences in uNK cell levels or gene expression between Hom-P and Het-U placentae? Establishing this is critical to interpretation of the data. This could easily be addressed either by performing histological analysis and staining with available reagents that readily detect NK cells. Alternatively this finding should be discussed in the text.

We assessed the abundance of uterine NK cells using the lectin, *Dolichos biflorus agglutinin* (DBA) and found no difference between Hom-P and Het-U placentas (see below). This data is described in the Results (see below) and shown in new Figure 6—figure supplement 7 of the revised manuscript.

“Granzymes, as well as perforin-1 (which was also differentially expressed in the Hom-P versus Het-U placenta; Figure 6—source data 1) are enriched in uterine natural killer cells. However, we found no significant difference in uterine natural killer cell abundance between Hom-P and Het-U placentas (Figure 6—figure supplement 3)”.